# Richter Syndrome: From Molecular Pathogenesis to Druggable Targets

**DOI:** 10.3390/cancers14194644

**Published:** 2022-09-24

**Authors:** Samir Mouhssine, Gianluca Gaidano

**Affiliations:** Division of Hematology, Department of Translational Medicine, Università del Piemonte Orientale and Azienda Ospedaliero-Universitaria Maggiore della Carità, 28100 Novara, Italy

**Keywords:** Richter syndrome, chronic lymphocytic lymphoma, pathogenesis, genetic lesions, targeted therapy

## Abstract

**Simple Summary:**

Chronic lymphocytic leukemia represents the most frequent leukemia in adults and evolves from an indolent phase to an aggressive lymphoma in 5–10% of the cases. Such clinico-pathologic transformation is known as Richter syndrome and represents a major unmet medical need because of treatment refractoriness, lack of targeted therapeutic strategies and unsatisfactory survival rates. Molecular investigations by innovative approaches have clarified in detail the pathogenesis of Richter syndrome and have led to the identification of the genetic and biologic changes associated with its evolution from indolent to aggressive disease. Knowledge of the molecular profile of Richter syndrome has revealed several molecular targets that may be exploited for devising novel therapeutic strategies, both with small molecules acting as pathway inhibitors and with monoclonal antibodies, that also include drug immunoconjugates. Clinical trials with these novel medicines are ongoing and may pave the way to a precision medicine approach to Richter syndrome.

**Abstract:**

Richter syndrome (RS) represents the occurrence of an aggressive lymphoma, most commonly diffuse large B-cell lymphoma (DLBCL), in patients with chronic lymphocytic leukemia (CLL). Most cases of RS originate from the direct transformation of CLL, whereas 20% are de novo DLBCL arising as secondary malignancies. Multiple molecular mechanisms contribute to RS pathogenesis. B-cell receptor (BCR) overreactivity to multiple autoantigens is due to frequent stereotyped BCR configuration. Genetic lesions of TP53, CDKN2A, NOTCH1 and c-MYC deregulate DNA damage response, tumor suppression, apoptosis, cell cycle and proliferation. Hyperactivation of Akt and NOTCH1 signaling also plays a role. Altered expression of PD-1/PD-L1 and of other immune checkpoints leads to RS resistance to cytotoxicity exerted by T-cells. The molecular features of RS provide vulnerabilities for therapy. Targeting BCR signaling with noncovalent BTK inhibitors shows encouraging results, as does the combination of BCL2 inhibitors with chemoimmunotherapy. The association of immune checkpoint inhibitors with BCL2 inhibitors and anti-CD20 monoclonal antibodies is explored in early phase clinical trials with promising results. The development of patient-derived xenograft mice models reveals new molecular targets for RS, exemplified by ROR1. Although RS still represents an unmet medical need, understanding its biology is opening new avenues for precision medicine therapy.

## 1. Definition of Richter Syndrome

Richter Syndrome (RS) was reported for the first time by Maurice N. Richter as “reticular cell sarcoma” in 1928 [1]. Currently, according to the World Health Organization (WHO) classification of Tumours of Haematopoietic and Lymphoid Tissues, RS is defined as the occurrence of an aggressive lymphoma in patients with a previous or concomitant diagnosis of chronic lymphocytic leukemia (CLL) or small lymphocytic lymphoma (SLL) [2]. RS is currently divided into two recognized pathological variants: diffuse large B-cell lymphoma (DLBCL) variant, with confluent sheets of large neoplastic post-germinal center B lymphocytes, and Hodgkin lymphoma (HL) variant [2,3].

The neoplastic cells of DLBCL-type RS express CD20, and less commonly CD5 and CD23 [2]. PD-1 expression is documented in up to 80% of DLBCL-type RS, whereas in de novo DLBCL, this marker is poorly expressed. Another difference between DLBCL-type RS and de novo DLBCL is the low rate of BCL2 genetic lesions, compared with the prevalence of BCL2 translocations and somatic mutations that are commonly found in de novo DLBCL [4,5,6]. The analysis of immunoglobulin genes has shown that ~80% of the cases of DLBCL-type RS are clonally related to the CLL phase, thus documenting that this histologic shift is a true transformation event from the previous indolent phase. However, a minority (~20%) of DLBCL-type RS cases are characterized by a rearrangement of immunoglobulin genes that is distinct from that of the CLL phase, documenting a clonally unrelated origin of RS [3].

The HL variant is similar to its de novo counterpart, with the presence of Hodgkin and Reed–Sternberg cells in a typical background of reactive T cells, epithelioid histiocytes, eosinophils and plasma cells or, eventually, interspersed in a background of CLL cells [7,8,9]. Hodgkin and Reed–Sternberg cells are characterized by a CD30+/CD15+/CD20− immunophenotype and are often found to be EBV positive [3,9].

## 2. Epidemiology

CLL is the most frequent leukemia in adults, with an incidence of 4.7/100,000 per year in the US [10]. In the chemo-immunotherapy (CIT) era, data mainly based on retrospective studies showed an incidence of RS transformation ranging from 1 to 7% [11]. Consistently, the CALGB 9011 clinical trial demonstrated a RS transformation rate of ~7% in treatment-naïve CLL patients after at least 15 years after treatment with fludarabine or chlorambucil [12]. Remarkably, in the CLL8 trial evaluating the effect of fludarabine and cyclophosphamide with or without the anti-CD20 monoclonal antibody (mAb) rituximab, the use of rituximab proved to be a protective factor against RS transformation, leading to lower rates of progression to RS in patients receiving rituximab [13]. A retrospective study suggested a potential role played by prolymphocytes in RS development, underlining that deaths due to RS were significantly more common in CLL patients who had ≥10% circulating prolymphocytes [14].

More recently, a pooled analysis of the German CLL Study Group (GCLLSG) considering frontline treatment trials with both CIT and pathway inhibitors, including Bruton tyrosine kinase (BTK) and BCL2 inhibitors, has documented a 3% prevalence of RS transformation among 2975 CLL patients monitored after their enrolment in clinical trials, recruited from 1999 to 2016, with a median observation time of 53 months [15]. Data from the Surveillance, Epidemiology and End Results (SEER) database of CLL patients diagnosed between 2000 and 2016 have documented that the incidence of RS transformation was 0.7% [16].

The issue of RS epidemiology in the era of novel agents has been partially answered by results collected from the first clinical trials with pathway inhibitors: in first-line treatment, novel agents showed RS transformation rates comparable to those of the CIT era, suggesting that pathway inhibitors are neither harmful nor fully protective [17,18]. Among relapsed/refractory (R/R) CLL patients treated with novel agents, the RS incidence was higher than the overall incidence of RS in the CIT era, probably due to the biological behavior and genetic profile of R/R CLL [19,20,21,22]. Because pathway inhibitors have been approved for the treatment of CLL relatively recently (the first BTK inhibitor, ibrutinib, was approved in 2014), further investigation is needed to assess the RS transformation rate with these novel agents in a real-life setting. Currently available data on RS frequency in cases treated with pathway inhibitors are summarized in Table 1.

## 3. Molecular Pathways in RS

Several molecular alterations associated with DLBCL-type RS have been described, whereas the development of HL-type RS has been studied less extensively and is thought to be similar to that of de novo HL and possibly linked to EBV-mediated immunosuppression, thus favoring CLL progression to HL [7,31,32].

*Genetic lesions of RS.* The pathogenesis of DLBCL-type RS is linked to the dysregulation of intracellular pathways involved in DNA damage response, tumor suppression, apoptosis, modulation of the cell cycle and proliferation. The main genetic lesions associated with DLBCL-type RS are represented by somatic mutations or disruptions of the TP53, CDKN2A, NOTCH1 and c-MYC genes (Figure 1) [2,5,6,7,33,34].

TP53 encodes for one of the main regulators of the DNA-damage-response pathway, and its disruption, generally acquired at the time of transformation, leads to the chemorefractoriness characteristic of RS and, therefore, favors the positive selection and expansion of mutated tumor cells [34,35]. Disruption of TP53 by mutation and/or deletion has been documented in a large fraction of DLBCL-type RS, including 60 to 80% of clonally related RS cases, which represent the overwhelming majority of RS events, and in 20% of clonally unrelated RS cases [6]. The high recurrence of TP53 disruption explains, at least in part, the frequency of chemorefractoriness in this condition.

CDKN2A deletion occurs in ~30% of DLBCL-type RS and is commonly acquired at the time of transformation [5]. The CDKN2A gene is responsible for the negative regulation of the G1 to S transition of the cell cycle and for the activation of the p53 transcriptional program through its transcripts p16INK4A and p14ARF, respectively, leading to tumor suppression [5,34,36]. Importantly, both positive and negative cell cycle regulators are induced by B-cell receptor (BCR) signaling in murine models. On these grounds, the concomitant loss of the negative cell cycle regulators TP53 and CDKN2A/B shifts BCR-dependent signaling toward the promotion of positive cell cycle regulators, leading to an aggressive proliferation compatible with RS transformation [37].

NOTCH1 encodes for a surface receptor that, after being triggered by a ligand belonging to the SERRATE/JAGGED or DELTA families, is cleaved by γ-secretase, migrates into the nucleus and activates the transcription of several genes involved in cell proliferation and survival [38]. At the time of diagnosis, the frequency of NOTCH1 mutational activation in RS is significantly higher compared to the frequency in CLL (31% vs. 8.3%, respectively) [39]. These data, together with the noticeably higher reported risk of developing DLBCL-type RS in NOTCH1 mutated CLL (45% at 15 years) vs. NOTCH1 wild type CLL (4.6% at 15 years), indicate that mutations of NOTCH1 are a significant risk factor for developing RS transformation [40].

The c-MYC proto-oncogene and its transcriptional product take part in many crucial cellular pathways, and a dysregulation of this network results in altered cell survival, proliferation, metabolism, self-renewal, and genomic instability [41]. The main genetic lesions deregulating c-MYC expression in DLBCL-type RS are represented by chromosomal translocations between the c-MYC locus and the IGHV regulatory regions, and gene amplifications as well as gain-of-function mutations of the c-MYC promoter. An alternative mechanism of c-MYC deregulation is represented by loss-of-function mutations of the MGA gene, which encodes for a protein inhibiting c-MYC heterodimerization with its partner MAX. Overall, by combining genetic lesions of c-MYC and MGA, ~40% of DLBCL-type RS cases harbor c-MYC deregulation [42].

The NF-κB pathway is also involved in RS pathogenesis [43]. TRAF3, a gene implicated in the negative regulation of signaling through the NF-κB and mitogen-activated protein kinase (MAPK) pathways, is disrupted by heterozygous deletions and frameshift mutations in a fraction of RS cases [43]. Inactivation of TRAF3 leads to NF-κB activation, promoting B cell survival and, in particular, enhancing the expression of c-MYC and PIM-2 [44]. PIM-2 maintains high levels of NF-κB, which are required for its antiapoptotic function. The pathogenetic role of PIM-2 in B-cell neoplasia is documented by its overexpression, translocation, or amplification in a fraction of B-cell lymphomas [43,44,45,46].

Other molecular alterations detected in DLBCL-type RS include (i) overexpression and amplification of PTPN11, a positive regulator of the MAPK-RAS-ERK signaling pathway; (ii) deletion of the SETD2 histone methyltransferase, which plays a major role in chromatin epigenetic remodeling; and (iii) disrupting mutations of the tumor suppressor gene PTPRD, which encodes for a receptor-type protein, tyrosine phosphatase, regulating cell growth and found to be inactivated also in other types of B-cell neoplasia and solid cancers [43,47,48,49,50].

*Modification of immune regulators in RS.* Recent studies have underlined the importance of immune checkpoints and of the tumor microenvironment in lymphatic tissues of DLBCL-type RS [4,51]. The main immune checkpoints involved in RS are PD-1, LAG3 and TIGIT.

PD-1 is a T cell surface molecule which stimulates effector T-cell apoptosis and Treg survival through its binding with the PD-L1 ligand, which is expressed mainly on the surface of antigen-presenting cells (APC), such as macrophages, B cells and dendritic cells (DC) [52]. Augmented levels of PD-1 in RS cells and enhanced expression of PD-L1 in histiocytes and dendritic cells of the RS microenvironment have been reported [4,51]. Altered expression of the PD-1/PD-L1 axis leads to RS tumor-cell resistance to the cytotoxicity exerted by T cells [52].

LAG3 is a T cell surface protein with a structure similar to the T helper antigen CD4, and its main ligand is the major histocompatibility complex (MHC) class II, typically expressed by APC. Additional ligands have been identified, namely Galectin-3 (Gal-3), which is expressed by several cell types [53]. The interaction between LAG3 and its ligands promotes tumor escape from apoptosis through the recruitment of tumor-specific CD4+ T cells (through the interaction with MHC class II) and the inhibition of CD8+ T cells’ cytotoxic function by Gal-3 binding [53]. Higher LAG3 levels have been observed in RS neoplastic and tumor-infiltrating lymphocytes, suggesting its potential role in promoting tumor immune escape and neoplastic cell survival [54].

TIGIT, expressed on normal T and NK cells and overexpressed in RS, is capable of immune suppression as a consequence of its binding with the CD155 ligand, exposed on the cell membrane of various cell types, such as dendritic cells, T cells, B cells, and macrophages [54,55]. The mechanism of action of TIGIT is supposed to be linked to the transduction of immune-suppressive stimuli on T and NK cells and to the promotion of tolerogenic DC that downregulate T cell responses [55]. The finding that immune checkpoints are overexpressed in RS cells and tumor-infiltrating lymphocytes suggests the potential role of these molecules in the promotion of a permissive immune microenvironment, resulting in immune suppression and tumor escape.

*The BCR pathway in RS.* Several BCR pathway alterations related to RS transformation have been documented. The BCR is a transmembrane complex expressed in B cells, composed by a surface immunoglobulin linked to a signal transduction subunit and responsible for antigen recognition and B cell activation (Figure 1) [56].

The variable part of the BCR IGHV subunit is characterized by a molecular pattern typical of mature B cells, the VDJ rearrangement, which causes a considerable diversity across the BCR expressed by different B cell clones. Approximately thirty percent of CLL patients carry stereotyped BCR, which are characterized by almost identical VDJ rearrangement across patients and are groupable in well-codified subsets identified by progressive numbers [57,58].

CLL patients carrying BCR subset 8 (characterized by IGHV4-39/IGHD6-13/IGHJ5 rearrangement) display a significantly higher risk of developing DLBCL-type RS, especially in combination with NOTCH1 mutations. [59,60]. From a mechanistic perspective, CLL cells harboring BCR subset 8 tend to overreact to multiple autoantigens and immune stimuli derived from the microenvironment (Figure 1). The propensity of these cells to undergo RS transformation can be explained by this promiscuous antigen reactivity [61].

In the BCR signal transduction, a key role is played by Bruton tyrosine kinase (BTK) and phosphatidylinositol-3 kinase (PI3K) [62]. BTK is phosphorylated subsequently to BCR stimulation and leads to the activation of phospholipase C gamma 2 (PLCγ2), causing calcium mobilization and activation of cell survival, proliferation and differentiation pathways, including MAPK and NF-κB signaling (Figure 1) [63]. PI3K, which is responsible for the activation of the serine/threonine kinase Akt and for the delta isoform of protein kinase C (PKC), was observed to be constitutively active in CLL patients, resulting in an enhanced anti-apoptotic effect [64].

*Akt signaling in RS.* Akt takes part in cell-survival signaling through mTOR and is constitutively active in high-risk CLL (i.e., TP53 or NOTCH1 mutated CLL) and in >50% of cases of RS [64,65]. In an Eµ-TCL1 murine model of CLL with constitutively active Akt alleles in B cells, the excessive Akt activation led to an aggressive DLBCL-type lymphoma with histological and biological features coherent with human RS [65]. Additionally, this murine model enlightened the correlation between hyperactivation of Akt and NOTCH1 signaling, since mice with constitutively active Akt alleles presented an expansion of CD4+ T cells expressing the NOTCH1 ligand DLL1 in the tumor microenvironment, implying a higher engagement of NOTCH1 by its ligands in neoplastic cells [65].

## 4. Molecular History of Clonally Related DLBCL-Type RS

Recently, an analysis performed on longitudinal tumor samples from CLL patients developing clonally related DLBCL-type RS has underlined the presence of dormant seeds of RS already at the time of CLL diagnosis [66]. The samples were obtained before and after treatment with chemotherapy, immunotherapy, and novel agents (mainly BCR pathway inhibitors), and were analyzed with bulk whole-genome, epigenome and transcriptome sequencing. A very early diversification of CLL was shown to lead to the emergence of neoplastic cells with RS features up to 19 years before the clinical and histological evidence of transformation [66]. These early clones displayed fully assembled genomic, immunogenetic and transcriptomic profiles of RS already at CLL diagnosis.

Moreover, RS tumor cells showed an activation of the oxidative phosphorylation (OXPHOS) pathway and, on the other side, a downregulation of BCR signaling [66]. The OXPHOS^high^–BCR^low^ transcriptional axis could partly explain the resistance and rapid selection of DLBCL-type RS cells observed after treatment with BCR inhibitors [67,68,69]. In this regard, OXPHOS inhibition has been tested in vitro, demonstrating an activity against the proliferation of DLBCL-type RS cells, a finding worth exploring in future therapeutic strategies [66].

## 5. Molecular Differences between Clonally Related and Clonally Unrelated DLBCL-Type RS

At least 80% of DLBCL-type RS cases carry the same IGHV rearrangement in both CLL and RS cells and thus originate from the direct transformation of the pre-existent CLL clone [3,34]. Conversely, a minority of DLBCL-type RS cases (approximately 20%) carry a different IGHV rearrangement compared to the CLL phase, and are therefore de novo DLBCL arising in the context of a condition, i.e., CLL, which predisposes to secondary lymphoid malignancies [3,34,70]. The distinction between clonally related and clonally unrelated DLBCL-type RS is clinically relevant, since clonally unrelated cases display a better prognosis that is overall overlapping with that of DLBCL arising de novo in patients without CLL [6,71].

The molecular pathogenesis of clonally unrelated DLBCL-type RS has been poorly understood for a long time. A very recent study exploited ultra-deep NGS to interrogate the timing of emergence and the genetic profile of this rare disorder [72]. The investigation of IGHV gene rearrangements of clonally unrelated DLBCL-type RS clarified that the RS clone was not present either at the time of CLL diagnosis or later in the clinical course of the indolent phase of the disease, even with the sensitivity of 10-6 allowed by ultra-deep NGS [72]. Therefore, clonally unrelated DLBCL-type RS unequivocally arises as a de novo DLBCL emerging as a secondary tumor whose development is favored by the deep immune suppression of the CLL host [70].

Mutational analysis performed by CAPP-Seq with an ample panel of genes affected in B-cell neoplasia revealed that genetic lesions of clonally unrelated DLBCL-type RS predominantly involve tumor suppressor genes implicated in the DNA damage response (TP53, ATM) as well as genes regulating cell cycle and proliferation (NOTCH1, ID3, and MYC) [72]. Remarkably, genes that are recurrently affected in DLBCL without a previous CLL phase, namely, KMT2D, CREBBP, EP300 and TNFAIP3, and genes implicated in BCR signaling are not affected in clonally unrelated DLBCL-type RS [72]. On these grounds, the genetic profile of clonally unrelated DLBCL-type RS is reminiscent, at least in part, of the mutational spectrum associated with clonally related DLBCL-type RS. The fact that both clonally unrelated and clonally related DLBCL-type RS share the involvement of a similar panel of genes despite their different histogenetic origins may reflect commonalities in the degree of immune suppression and/or in the lymph node microenvironment in both conditions. Also, the previous exposure of the host to antineoplastic agents might exert a role favoring the accumulation of specific genetic lesions in both clonally unrelated and clonally related DLBCL-type RS.

From a clinical standpoint, this novel information on clonally unrelated DLBCL-type RS is of value. On the one side, in fact, the molecular vulnerabilities of both clonally related and clonally unrelated DLBCL-type RS have a certain degree of overlap and might be exploited as common targets with innovative agents [72]. On the other side, the high frequency of TP53 disruption in both clonally related and clonally unrelated DLBCL-type RS mandates the analysis of this genetic lesion in the RS sample, since TP53 disruption is a consolidated predictor of chemorefractoriness in the context of both CLL and DLBCL arising de novo [73,74]. Finally, because the treatment of clonally related versus clonally unrelated DLBCL-type RS differs, at least for what concerns the indication to hematopoietic stem cell transplant, whenever possible, clinicians should implement strategies aimed at defining the clonal relationship between CLL and DLBCL-type RS [34,75]. From a practical point of view, it might be cumbersome to have the availability of both samples for genomic analysis. However, interpolation between PD-1 expression and immunoglobulin genes analyses showed a 90% correlation of PD-1 expression with clonally related DLBCL-type RS defined in molecular terms; consequently, PD-1 expression could represent a surrogate for defining the clonal relation between CLL and DLBCL-type RS [4].

## 6. Druggable Targets for the Treatment of DLBCL-Type RS

The understanding of the molecular pathways involved in RS pathogenesis has led to the identification of novel druggable targets, namely BTK, BCL2, PD-1 and PD-L1 (Figure 2).

*Basic concepts in the treatment of RS.* DLBCL-type RS is characterized by an aggressive and particularly chemorefractory behavior, partly due to the genetic features of the disease. Furthermore, RS patients commonly suffer from poor performance status, impaired bone marrow function or immunodeficiency, displaying a frailty profile that represents a predictor of low survival rate after RS transformation [76]. The combination between chemorefractoriness, on one side, and frailty, on the other side, may be a possible cause of the dismal prognosis observed in RS treated with CIT because of the limited possibility to tolerate high-dose chemotherapy typical of frail and heavily pretreated patients. In a phase II trial, the R-CHOP (rituximab, cyclophosphamide, doxorubicin, vincristine, and prednisone) regimen, widely used in de novo DLBCL, showed an overall response rate (ORR) of 67% in RS patients, with a complete response (CR) of only 7%, a median overall survival (OS) of 21 months and a median progression-free survival (PFS) of 10 months [77]. The most common severe adverse events were hematological toxicity (grade ≥3 anemia, neutropenia and thrombocytopenia) and infections. Regimens alternative to R-CHOP have also been tested for RS, although none of them has demonstrated a clear superiority. A retrospective study assessed first-line R-EPOCH (rituximab, etoposide, prednisone, vincristine, cyclophosphamide, and doxorubicin) and resulted in a median OS of 5.9 months and underlined a poorer prognosis in patients with TP53 deletion or complex karyotype [78]. The hyper-CVXD (fractioned cyclophosphamide, vincristine, doxorubicin, and dexamethasone) regimen has been investigated in a phase II study and, despite the clinically significant CR rate of 38%, an important toxicity was reported, with an overall mortality rate of 20% [79]. A phase I/II study assessing safety and efficacy of the OFAR2 (oxaliplatin, fludarabine, ara-C and rituximab) regimen reported an ORR of 38.7%, with a CR of 6.5% and a median survival duration of 6.6 months [80]. The CHOP-OR phase II trial evaluated the CHOP-O (cyclophosphamide, doxorubicin, vincristine, prednisolone, with induction and maintenance with the anti-CD20 mAb ofatumumab) regimen in newly diagnosed RS, reporting an ORR of 46%, with a CR rate of 27%. The median PFS of 6.2 months and the median OS of 11.4 months did not demonstrate superiority compared to R-CHOP in terms of outcome [81]. These findings highlight the need for alternative therapeutic strategies in RS treatment.

*BTK inhibitors in RS.* Covalent BTK inhibitors challenge the Cys481 residue of BTK, which lies within the ATP-binding pocket of the kinase, resulting in a BCR pathway blockade (Figure 2) [82]. Ibrutinib, the first-in-class BTK inhibitor that binds covalently to Cys481, has shown a limited efficacy profile in DLBCL-type RS [83,84,85,86]. An analysis of the ACE-CL-001 study led to similar outcomes with acalabrutinib, a second generation covalent BTK inhibitor, with a CR of 8% and a median PFS of only 3 months [87]. The poor efficacy profile displayed by covalent BTK inhibitors may be explained, at least in part, by the development of BTK inhibitor-resistant clones, which harbor mutations of BTK affecting Cys481 (the most common is C481S) and gain-of-function mutations of PLCγ2 [82,88]. For this reason, a reversible noncovalent BTK inhibitor may be preferred. Initial results of the use of noncovalent inhibitors are coming from an ongoing phase I/II study (NCT03740529) evaluating the safety and efficacy of pirtobrutinib, a noncovalent BTK inhibitor, in previously treated DLBCL-type RS [89]. Pirtobrutinib showed encouraging results, with an ORR of 75% and few toxicity events in patients with a median number of two prior lines of therapy [89,90]. A promising reversible noncovalent BTK inhibitor is nemtabrutinib (ARQ 531), tested in murine models of CLL and aggressive B-cell lymphoma [91]. Nemtabrutinib demonstrated a better efficacy profile compared with ibrutinib in vivo, and two clinical trials involving RS patients are ongoing (NCT03162536 and NCT04728893) [91].

*BCL2 inhibitors in RS.* The BCL-2 family of proteins plays a central role in inhibition of apoptosis, leading to an enhanced tumor cell survival if overexpressed [92]. The BCL-2 inhibitor venetoclax has shown efficacy in CLL, independent of TP53 mutations and chromosome 17p deletion (Figure 2) [93,94]. DLBCL-type RS, as already mentioned, commonly harbors TP53 disruption, making venetoclax a potentially viable strategy for RS treatment. A phase 1 study (NCT01328626) attained a response rate of 43% in DLBCL-type RS treated with venetoclax monotherapy, but no cases of CR were reported [95]. More positive results have come from a phase II trial (NCT03054896), testing the combination of venetoclax with R-EPOCH in DLBCL-type RS, that reached an ORR of 62%, with a remarkable CR rate of 50% (13/26, while 11 of them achieved undetectable bone marrow minimal residual disease for CLL) [96]. The median PFS and median OS were 10.1 months and 19.6 months, respectively, and the grade ≥3 AEs were neutropenia, thrombocytopenia, and febrile neutropenia [96].

*Immune checkpoint inhibitors in RS.* PD-1 signaling can be targeted with mAbs that interfere with the PD-1/PD-L1 interaction (Figure 2) [94]. Pembrolizumab, an anti-PD-1 mAb, was evaluated in a phase II clinical trial (NCT02332980), in which DLBCL-type RS patients achieved an ORR of 40% and a median OS of 11 months [97]. The safety profile was acceptable, with only 20% of patients experiencing severe hematologic toxicity. These results were obtained in a non-homogeneous cohort of RS patients, in which 70% of cases were previously treated with ibrutinib, underlining the need for further studies in this treatment strategy [97]. Pembrolizumab monotherapy was assessed also in 21 R/R RS patients in the KEYNOTE-170 phase II study, which underlined poor results for R/R DLBCL-type RS; in particular, the ORR was 4.8% (only 1 patient), with no CR [98].

Combination therapies with immune checkpoint inhibitors seem to grant more positive results compared to monotherapy. Nivolumab, a PD-1 inhibitor, in combination with ibrutinib displayed an ORR of 65%, with a 10% CR rate, in a DLBCL-type RS cohort of a phase I/II study [99]. Encouraging results came from an ongoing phase II trial investigating the anti-PD-L1 mAb atezolizumab in combination with venetoclax and the anti-CD20 mAb obinutuzumab in treatment-naïve and R/R RS [100]. The initial results of the study reported an ORR of 100%, with a 71% CR in the treatment-naïve cohort, whereas only one patient was recruited in the R/R group [100].

## 7. Druggable Targets for the Treatment of HL-Type RS

Due to its similarity to de novo cases, HL-type RS is mainly treated with the ABVD (Doxorubicin, Bleomycin, Vinblastine, Dacarbazine) chemotherapy regimen, with a reported response rate of 40–60% and a median OS of ~4 years [101,102,103]. The main possible adverse event is the severe pulmonary toxicity caused by the exposure to bleomycin, which can be omitted after two cycles of ABVD if interim PET shows remission with no significant outcome variations [104].

Immune checkpoint inhibitors and brentuximab vedotin, an anti-CD30 conjugated with the antimicrotubular agent monomethyl auristatin E (MMAE), have proved to be effective in de novo R/R HL, but data on the activity on R/R HL-type RS are insufficient [105,106]. A case report showed encouraging results in R/R HL-type RS, with a successful outcome in a patient refractory to bendamustine, which at relapse was treated with brentuximab, vedotin and radiotherapy, and, later, with pembrolizumab [107]. Based on this evidence, investigations with these innovative drugs might be envisaged in HL-type RS.

## 8. Perspectives of New Potential Molecular Targets for RS

Recently, patient-derived xenograft (PDX) models for DLBCL-type RS have been developed in order to study human-like models of this rare disease, overcoming the limited number of patients [108]. Thanks to PDX models, promising pre-clinical results are being reached, paving the way for further clinical trials.

The receptor tyrosine kinase-like orphan receptor 1 (ROR1) is a surface receptor tyrosine kinase that, through the binding to proteins of the Wnt family, primarily Wnt5a, activates non-canonical WNT signaling, promoting cell survival, proliferation and migration events [109]. Despite its important role in embryogenesis, ROR1 is not expressed in the majority of normal adult tissues and cells, such as B cells, whereas it is overexpressed in many cancer cell types, including CLL and DLBCL-type RS [110,111,112,113]. The effect of the antibody drug conjugate (ADC) VLS-101, consisting of a mAb targeting ROR1 linked to the antimicrotubular agent MMAE, has been assessed in a xenograft model [114]. When VLS-101 binds to ROR1 on DLBCL-type RS cells, the complex is internalized by the cell and included in lysosomes, where, via proteolytic cleavage, MMAE is released and can fulfill its cytotoxic function, blocking cell cycle progression and inducing apoptosis in the neoplastic cell (Figure 2) [114]. This cellular effect translates into reduction of tumor burden and into prolonged mice survival, with high selectivity and no toxicities recorded in treated mice [114]. Based on these encouraging findings, a phase I clinical trial testing VLS-101 in hematological malignancies, including RS, has been recently started (NCT03833180). Other ADCs are giving early promising results in PDX models, such as immunoconjugates linked to the cytotoxic payload amanitin and directed against CD37, an antigen expressed on the cellular membrane of DLBCL-type RS cells [115].

Due to the constitutive activation of the PI3K/Akt pathway in DLBCL-type RS, a targeted therapeutic strategy might be represented by PI3K inhibition (Figure 2). Although the PI3Kδ inhibitor idelalisib showed some clinical effect in DLBCL-type RS in a retrospective study, the toxicities linked to this drug represent a major obstacle to treatment, as extensively reported in CLL patients [116,117,118]. More recently, in RS-PDX models, the PI3Kδ/γ inhibitor duvelisib, combined with venetoclax, showed encouraging efficacy and selectivity, leading to an increased apoptosis rate in RS cells and a significant reduction of tumor volume coupled to a prolonged survival in treated mice [119]. This synergistic effect is probably obtained through the cross-talk between PI3K and BCL2 pathways, thanks to the activation of GSK3β by the inhibition of Akt promoted by duvelisib, resulting in reduced levels of the anti-apoptotic protein Mcl-1, which, in turn, leads to sensitization to venetoclax [119,120]. These encouraging results led to the design of a phase I trial with duvelisib and the anti-PD-1 mAb nivolumab, which is still ongoing (NCT03892044). A potential target can also be represented by Akt, tested only in CLL with the allosteric inhibitor MK-2206, in combination with a BR (rituximab plus bendamustine) regimen (Figure 2) [121]. MK-2206 obtained promising results in terms of safety and efficacy, but, due to the decision of the sponsor to stop its development, further studies with alternative Akt inhibitors deserve to be undertaken in RS.

Another possible target for RS therapy could be represented by exportin 1 (XPO1), a key mediator of the nuclear export of macromolecules [122]. Recently, the XPO1 gene has been found to be frequently overexpressed or mutated in CLL and affected by gain-of- function point mutations in 25–35% of RS, resulting in an unbalance between pro-oncogenic and antioncogenic proteins (i.e., p53) (Figure 1 and Figure 2) [123,124]. Selinexor, a XPO1 inhibitor, was evaluated in a phase I study that reported an ORR of 33% in R/R DLBCL-type RS, with an acceptable safety profile [125]. Unfortunately, the phase II trial (NCT02138786) assessing selinexor monotherapy in RS was closed due to lack of efficacy, but strategies involving combination therapies with selinexor or other XPO1 inhibitors might be explored.

Since TP53 is commonly disrupted in RS, an intriguing potential therapeutic approach might include TP53-pathway-restoring drugs under development [126]. APR-017 and APR-246 are first-in-class drugs, capable of binding to cysteine residues in the DNA binding domain of p53, restoring its physiological conformation in TP53 mutated in vitro models of CLL [126,127]. In TP53-deleted CLL, evidence of the effectiveness of RG7388, an inhibitor of the downregulating p53 ligand MDM2 (an E3 ubiquitin ligase), has been documented in vitro (Figure 2) [128]. Because these compounds are not yet included in clinical trials for RS patients, further investigation into this therapeutic strategy is needed.

Lastly, the inhibition of cyclin-dependent kinase 4 and 6 (CDK4/6), involved in the positive regulation of the cell cycle, might represent a way to overcome CDKN2A/B deletion in RS. The CDK4/6 inhibitor palbociclib showed activity both in vitro and in vivo in preclinical models of RS, with a synergistic effect with BCR inhibitors [37,129]. Ongoing clinical trials in RS are depicted in Table 2 (https://clinicaltrials.gov/, last access on 3 August 2022).

## 9. Conclusions

In clinical practice, RS still represents a major unmet medical need, and most patients die because of a lack of dedicated treatments and/or because of frailty due to previous exposure to multiple lines of ineffective therapy. The molecular pathogenesis of RS, in particular of DLBCL-type RS, has been elucidated to a certain extent, revealing druggable vulnerabilities both in the tumor clone and in the RS microenvironment. The availability of pathway inhibitors, namely BCR and BCL2 inhibitors, coupled with the development of innovative mAbs with different modes of action, is substantially changing the therapeutic landscape of RS. These novel medicines are currently being tested in clinical trials with or without chemoimmunotherapy, and a few promising results are emerging. The development of PDX models of RS may represent a further tool for testing novel drug combinations that, if successful in these pre-clinical models, may then be exported to clinical trials in RS patients. A precision medicine approach to RS may thus be considered for the future and will require a detailed characterization of the clonal relationship, of the genetic profile and of the expression pattern of RS cells in individual patients.

## Figures and Tables

**Figure 1 cancers-14-04644-f001:**
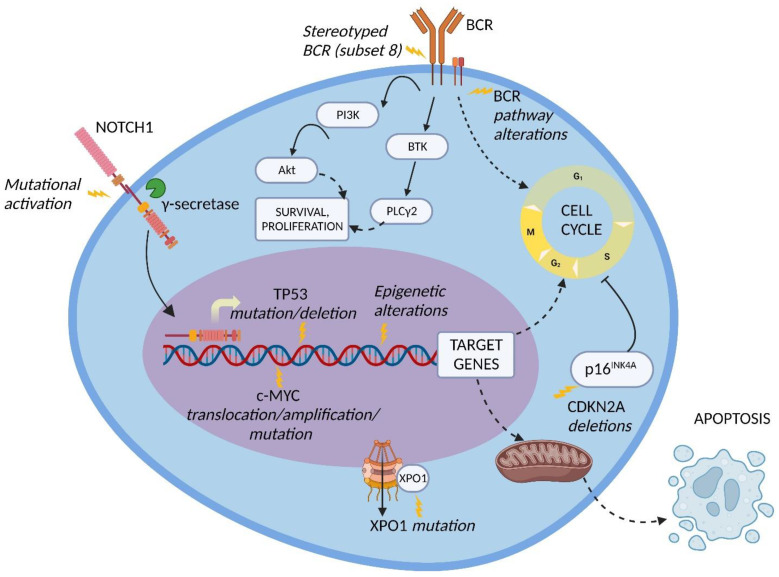
Molecular pathway alterations in DLBCL-type RS. The pathogenesis of DLBCL-type RS is due to the dysregulation of multiple molecular pathways due to genetic lesions of proto-oncogenes and tumor suppressor genes, stereotyped B-cell receptor (BCR) configuration, and BCR signaling alterations; these lead to the enhanced cell survival and proliferation typical of DLBCL-type RS cells. Inhibition of apoptosis may also be involved. BTK, Bruton tyrosine kinase; PI3K, phosphatidylinositol-3 kinase; PLCγ2, phospholipase C gamma 2; XPO1, exportin 1. Image created with Biorender.com (accessed on 4 August 2022).

**Figure 2 cancers-14-04644-f002:**
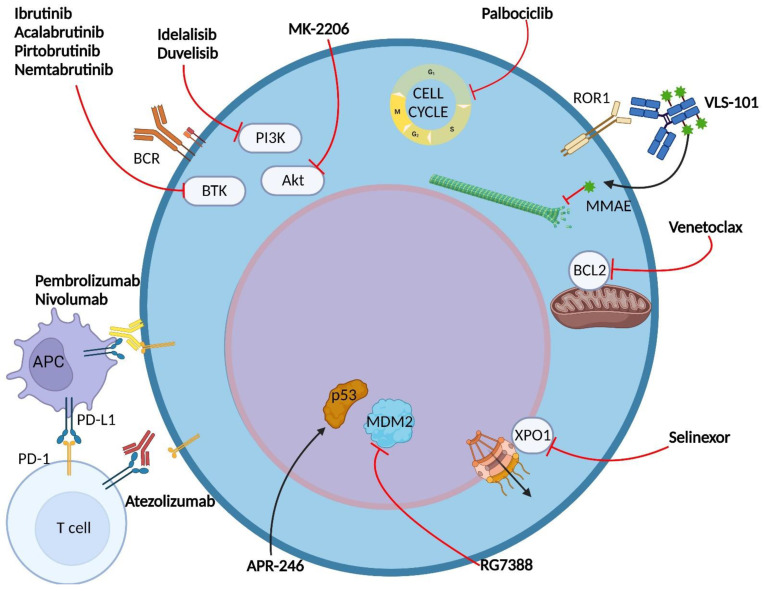
Drugs under development for the targeted therapy of DLBCL-type RS. The figure represents the main drugs and targets under development for the molecular treatment of DLBCL-type RS. The agents depicted here can be divided into small molecules and monoclonal antibodies (mAb). Small molecules target BTK, PI3K, Akt, CDK4/6, BCL2, XPO-1, MDM2 and p53. mAb include immune checkpoint inhibitors and antibody drug conjugates (ADC). The anti-ROR1 ADC VLS-101 interrupts the microtubule polymerization through the release of its payload monomethyl auristatin E (MMAE) in the cytoplasm of the RS cell. APC, antigen presenting cell; BCR, B cell receptor; BTK, Bruton tyrosine kinase; PI3K, phosphatidylinositol-3 kinase; XPO1, exportin 1; CDK4/6, cyclin-dependent kinase 4 and 6; ROR1, receptor tyrosine kinase-like orphan receptor 1. Image created with Biorender.com (accessed on 4 August 2022).

**Table 1 cancers-14-04644-t001:** RS frequency in patients treated with therapeutic regimens based on pathway inhibitors.

Number of CLL Patients	Study Population	Treatment	RS Prevalence (%)	Reference
391	Relapsed	Ibrutinib, ofatumumab	1	Byrd, 2014 [23]
29	Progressive untreated	Ibrutinib	3	O’Brien, 2014 [24]
194	R/R	Venetoclax-rituximab	3	Seymour, 2018 [19]
127	R/R	Ibrutinib	5	Jain, 2015 [25]
84	17p deleted or ≥65 years	Ibrutinib	6	Ahn, 2017 [21]
358	Treatment-naïve	Acalabrutinib, Obinutuzumab	2	Sharman, 2020 [18]
51	17p deleted	Ibrutinib	6	Farooqui, 2015 [26]
178	BCRi treated	Ibrutinib, idelalisib	7	Mato, 2016 [27]
113	Treatment-naïve	Ibrutinib-obinutuzumab	0	Moreno, 2019 [17]
85	R/R	Ibrutinib	8	Byrd, 2013 [28]
116	R/R	Venetoclax	16	Roberts, 2016 [29]
67	R/R, 17p deleted	Venetoclax	25	Anderson, 2017 [30]
2975	R/R	B, F, C, Clb, rituximab, obinutuzumab, ibrutinib, venetoclax	3	Al-Sawaf, 2021 [15]
195	R/R	Ibrutinib	10	Munir, 2019 [20]

Abbreviations: RS, Richter Syndrome; R/R, relapsed/refractory; BCRi, B-cell receptor pathway inhibitors; B, bendamustine; F, fludarabine; C, cyclophosphamide; Clb, chlorambucil.

**Table 2 cancers-14-04644-t002:** Summary of ongoing trials in RS.

Intervention	Phase	Main Target	NCT Number
VR-EPOCH/VR-CHOP	II	BCL2	NCT03054896
Obinutuzumab + rituximab + venetoclax	II	BCL2	NCT04939363
R-CHOP + blinatumomab	II	CD19	NCT03931642
Epcoritamab	I/II	CD20	NCT04623541
Polatuzumab vedotin + R-EPCH	II	CD79b	NCT04679012
Duvelisib + venetoclax	I/II	PI3K, BCL2	NCT03534323
Zanubrutinib + tislelizumab	II	BTK, PD-1	NCT04271956
Acalabrutinib + durvalumab + venetoclax	II	BTK, PD-L1, BCL2	NCT05388006
Duvelisib + nivolumab	I	PI3K, PD-1	NCT03892044
VIP152	I	CDK9	NCT04978779
Atezolizumab + Obinutuzumab + venetoclax	II	PD-L1, BCL2	NCT02846623
Acalabrutinib	I/II	BTK	NCT02029443
Copanlisib + nivolumab	I	PI3K, PD-1	NCT03884998
Obinutuzumab + atezolizumab + venetoclax	II	PD-L1, BCL2	NCT04082897
Cosibelimab ± ublituximab and bendamustine	I	PD-L1	NCT03778073
MOR00208 ± lenalidomide or ibrutinib	II	CD19, BTK	NCT02005289
Polatuzumab vedotin	I/II	CD79b	NCT04491370
Pembrolizumab + acalabrutinib	I/II	PD-1, BTK	NCT02362035
Pevonedistat	I	NAE, BTK	NCT03479268
Pembrolizumab ± idelalisib or ibrutinib	II	PD-1, PI3K, BTK	NCT02332980
LP-118	I	BCL2	NCT04771572
TG-1801 ± ublituximab	I	CD47, CD19	NCT04806035
NX-1607	I	CBL-B	NCT05107674
Nemtabrutinib	I/II	BTK	NCT03162536
VLS-101	I	BTK	NCT03833180
DTRMWXHS-12+everolimus ± pomalidomide	II	BTK, mTOR	NCT04305444
HMPL-760	I	BTK	NCT05176691
ALX148 + rituximab + lenalidomide	I/II	CD47	NCT05025800

Abbreviations: VR-EPOCH, venetoclax + rituximab + etoposide + prednisone + vincristine + cyclophosphamide + doxorubicin; R-CHOP, rituximab + cyclophosphamide + doxorubicin + vincristine + prednisone; VR-CHOP, venetoclax + R-CHOP; R-EPCH, rituximab + etoposide + prednisone + cyclophosphamide + hydroxydaunorubicin; NAE, NEDD8-activating enzyme; BTK, Bruton tyrosine kinase; PI3K, phosphatidylinositol-3 kinase; PD-1, programmed death 1; PD-L1, programmed death ligand 1; CBL-B, Casitas B-lineage lymphoma.

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
