# Peer review of "Richter Syndrome: From Molecular Pathogenesis to Druggable Targets"

_cancers, 2022, doi:10.3390/cancers14194644_

Round 1

Reviewer 1 Report

The molecular pathogensis, novel molecular therapeutic targets, and novel experimental models for exploring new potential new molecular targets in RS has been reviewed in the manuscript.

My only comment is regarding figure 1. Altered molecular pathways involved in pathogensis of RS (enhanced susrvival and proliferation) are presented, and for me it is nor clear why apoptotic cell (in the lower right corner)  is included in the figure?

Author Response

Reviewer 1: The molecular pathogensis, novel molecular therapeutic targets, and novel experimental models for exploring new potential new molecular targets in RS has been reviewed in the manuscript.

My only comment is regarding figure 1. Altered molecular pathways involved in pathogensis of RS (enhanced susrvival and proliferation) are presented, and for me it is nor clear why apoptotic cell (in the lower right corner)  is included in the figure?

We thank the Reviewer for his/her constructive comment. Since apoptosis is a key physiological process in normal cells and it is found to be inhibited in DLBCL-type RS, we included the apoptotic cell in figure 1 and we specified this concept in the figure legend.

Reviewer 2 Report

The MS is an updated comprehensive overview that focuses on drugs in development, with nicely designed figures and detailed information about ongoing clinical trials. I enjoyed reading this MS. Undoubtedly, this article will be of interest to clinical hematologists, and is highly recommended for medical hematology students. 

Author Response

Reviewer 2: The MS is an updated comprehensive overview that focuses on drugs in development, with nicely designed figures and detailed information about ongoing clinical trials. I enjoyed reading this MS. Undoubtedly, this article will be of interest to clinical hematologists, and is highly recommended for medical hematology students. 

We give thanks to the Reviewer for his/her appreciative words in regard of our work.

Reviewer 3 Report

The Review entitled "Richter syndrome: from molecular pathogenesis to druggable targets" by Mouhssine and Gaidano is well written and organized. However, my major concern regards the lack of novelty since similar reviews have been recently published, including Condoluci and Rossi in Current Oncology Reports 2021 and Condoluci and Rossi in Frontiers in Oncology 2022. This review indeed mostly overlaps these cited reviews, both as organization (Definition of Richter syndrome, Epidemiology and so on) and text, including Table 2 that is similar to Table 2 of the Review published in Frontiers in Oncology.  

Author Response

Reviewer 3: The Review entitled "Richter syndrome: from molecular pathogenesis to druggable targets" by Mouhssine and Gaidano is well written and organized. However, my major concern regards the lack of novelty since similar reviews have been recently published, including Condoluci and Rossi in Current Oncology Reports 2021 and Condoluci and Rossi in Frontiers in Oncology 2022. This review indeed mostly overlaps these cited reviews, both as organization (Definition of Richter syndrome, Epidemiology and so on) and text, including Table 2 that is similar to Table 2 of the Review published in Frontiers in Oncology. 

We thank the Reviewer for his/her comments. In order to address the issue raised by the Reviewer we have added a novel paragraph that focuses on the newly reported data on the detection of early seeding of Richter transformation in the natural history of chronic lymphocytic leukemia (lines 230-247). Regarding Table 2 of our manuscript, it provides updated data in comparison to the Review published in Frontiers in Oncology.